# VMD–RP–CSRN Based Fault Diagnosis Method for Rolling Bearings

**Yuanyuan Jiang [1,2] and Jinyang Xie [3,*]**

1   School of Electrical and Information Engineering, Anhui University of Science and Technology,
    Huainan 232000, China
2   Institute of Environment-Friendly Materials and Occupational Health, Anhui University of Science and
    Technology, Wuhu 241003, China
3   School of Institute of Artificial Intelligence, Anhui University of Science and Technology,
    Huainan 232000, China
*   Correspondence: xjy19980920@163.com

**Abstract:** In response to the problems of low accuracy and poor noise immunity of the traditional fault diagnosis method for rolling bearing fault diagnosis due to the complex and variable operating conditions of rolling bearings and the large noise interference during bearing signal acquisition, a rolling bearing fault diagnosis model based on VMD–RP–CSRN is proposed. Firstly, the initial feature extraction of the bearing signal is carried out by variational modal decomposition (VMD), which is then converted into a two-dimensional image with fault features by recurrent plot (RP) coding, and then the feature images are input to a channel split residual network (CSRN) for feature extraction and fault classification. In order to verify the accuracy and noise immunity of the proposed method for the diagnosis of bearing faults under complex working conditions, experiments on the selection of parameters in the CSRN model were conducted on the bearing dataset of Jiangnan University, and experiments on the diagnosis of bearing faults under complex working conditions and noise immunity of CSRN were carried out and compared with other commonly used methods. The proposed bearing fault diagnosis method based on VMD–RP–CSRN combines VMD and RP to retain the fault features in the original signal to the maximum extent and stress the hidden features in the signal. The proposed channel split operation realizes the extraction of hidden features by selecting the main operating channel of the three-channel feature image, and makes more fault features participate in the feature extraction of the diagnosis model. The experimental results demonstrate that the proposed method is at least 1.2% better than the comparison method, and has better noise immunity. In addition, experiments on the fault diagnosis capability of the model with different data set sizes and the diagnosis of variable speed bearing data by the model show that the proposed method has better generalization performance and diagnosis capability.

**Keywords:** bearing fault diagnosis; variational modal decomposition; recurrent plot; channel split residual network

## 1. Introduction

A rolling bearing is the core part of rotating machinery. Its running condition is related to whether the rotating machinery can work safely and stably. In order to ensure the safety and stability of machinery operation, it is important to carry out fault diagnosis and online monitoring of rolling bearings [1]. The diagnostic accuracy of the traditional fault diagnosis method is very dependent on the effectiveness of the fault feature extraction. In the face of complex and changing working conditions, the feature extraction method is obviously insufficient for the extraction of fault features, which leads to the unsatisfactory diagnosis effect and poor noise immunity of the traditional fault diagnosis method for multi-condition bearing faults [2]. Therefore, it is important to study a fault diagnosis

method that can make up for the lack of feature extraction algorithm, has high diagnostic capability for bearing faults and has strong noise immunity.

Traditional fault diagnosis methods achieve the extraction of fault signals and fault diagnosis by performing time–frequency domain analysis on the original bearing fault signals. For example, the fault features are extracted by empirical modal decomposition (EMD) [3], ensemble empirical modal decomposition (EEMD) [4], singular value decomposition (SVD) [5], and wavelet transform (WT) [6], and then diagnosed by back propagation (BP) neural networks [7], support vector machines (SVM) [8], etc. Although these methods can usually carry out effective bearing fault diagnosis, they are not effective for fault diagnosis under complex operating conditions and noise interference, and have poor robustness.

With the development of deep learning, its powerful feature extraction capability has led to a new solution to the bearing fault diagnosis problem [9]: fault classification by encoding a one-dimensional bearing fault signal into a two-dimensional image and then feeding it into a convolutional neural network [10]. Commonly used data conversion methods include EMD binarization [11], G. Angular Difference Fields (GADF) [12], multiwavelet transform [13], Signal-to-Image Mapping (STIM) [14], etc., and have been used in bearings. In order to further improve the diagnostic accuracy of bearing faults, Che Changchang et al. [15] used the bearing data to construct grey-scale map fault samples and input them into a deep residual shrinkage network for fault diagnosis, which solved the degradation problem of the multilayer network model and improved the diagnostic accuracy by adding residual shrinkage blocks. This avoids feature loss and increases the accuracy. Dechen Yao et al. [16] added reverse residual blocks to the network to increase data dimensions before feature extraction, avoiding feature loss and improving accuracy. To address the problem of variable operating conditions, Ke Zhang et al. [17] proposed a multi-mode convolutional neural network, which used multiple parallel convolutional layers to extract fault features and then transformed the 1D data of rolling bearings under different frequency conversion conditions into 2D time–frequency gray scale maps by Continuous Wavelet Transform (CWT), and put them into a multi-mode convolutional neural network. When the bearings are operated under different loads, the same fault features under different loads are only different in terms of feature frequencies, which are difficult to distinguish effectively, and it is difficult for the above fault diagnosis methods to extract these features effectively, resulting in poor diagnostic accuracy of the diagnostic model.

In order to highlight the features in the bearing vibration signals, so that the diagnostic model can make a more accurate diagnosis of bearing faults, some scholars carry out a secondary extraction of bearing fault features after feature extraction, and select representative features as the fault classification basis. Jovan Gligorijevic et al. [18] decomposed the bearing vibration signal into several interested sub-bands through wavelet decomposition, and took the standard deviation of the obtained wavelet coefficient as the representative feature to realize the accurate diagnosis of bearing faults. Aleksandar Brkovic et al. [19] made wavelet decomposition of bearing vibration signals, extracted the standard deviation as a measure of average energy and the logarithmic energy entropy as a measure of disorder degree from the interested sub-bands as representative features, and used the scattering matrix to optimize their dimensions, achieving 100% diagnostic accuracy in the early bearing fault diagnosis.

VMD, proposed by Konstantin Dragomiretskiy in 2014, is an adaptive and completely non-recursive approach to modal variational and signal processing, which can adaptively match the optimal center frequency and finite bandwidth of each mode in the search and solution process according to the given number of modal decompositions, and can achieve the effective separation of the intrinsic modal components and the signal. The optimal solution of the variational problem is obtained by dividing the frequency domain. In the bearing fault diagnosis, Hongjiang Cui et al. [20] decomposed the bearing vibration signal into a series of intrinsic mode functions by VMD, then classified the fault features of maximum correlation kurtosis deconvolution, and obtained a better fault diagnosis accuracy. After

decomposing the original signal into mode components and dividing the mode matrix into sub-matrices, Chang Liu et al. [21] extracted the local feature information contained in each sub-matrix into singular value vectors using singular value decomposition, constructed the singular value vector matrix corresponding to the current fault state according to the position of each sub-matrix, and finally completed the identification and classification of fault types by the convolutional neural network.

In order to improve the diagnostic accuracy of bearing faults, and to give full play to the feature extraction algorithm's ability to extract bearing fault features and the convolutional neural network's powerful adaptive feature extraction ability for images, a new fault data processing method based on the combination of VMD feature extraction algorithm and recursive graph data coding method is proposed: VMD–RP, which converts bearing faults into two-dimensional after coding them by VMD–RP, and combines the channel split residual network (CSRN) to perform adaptive feature extraction on them to achieve fault classification. The main contributions of this paper are as following:

1.  Combine VMD feature extraction algorithm with RP image coding to transform feature extraction of fault data into two-dimensional images and enhance the correlation between time series data. On the premise of fully retaining the features contained in the original fault signal, the hidden features in the signal are mined through VMD and expressed through RP.
2.  Build the channel split mechanism, improve the residual network, make full use of the differences of features in different channels of two-dimensional images, and selectively highlight the channels, so as to fully express the hidden feature information in the channels and fully extract the hidden features of RP images.

## 2. VMD Principle

The VMD algorithm is an adaptive non-recursive modal decomposition method. The method uses the alternating direction multiplier algorithm to iterate sequentially to find the optimal solution of the constrained variational model, thus obtaining the intrinsic mode function (IMF) with $K$ central frequencies of $\omega_k$. The decomposition process of the VMD can be summarized as follows: Where $\mu_k$ and $\omega_k$ respectively represent each mode signal and the center frequency, $\alpha$ is the quadratic penalty factor, $\check{}$ is the Lagrange operator.

Step 1: Initialisation $\{\mu_k^1\}$, $\{\omega_k^1\}$, $\lambda^1$, $n \leftarrow 0$;

Step 2: Let $n = n + 1$ and $k = k + 1$ and update $\hat{\mu}_k^{n+1}$ and $\omega_k^{n+1}$ by Equations (1) and (2) respectively. Then stop iterating when $k = K$.

$$\hat{\mu}_k^{n+1}(\omega) = \frac{\hat{f}(\omega) - \sum_{i \neq k} \hat{\mu}_i^{n+1}(\omega) + \frac{\hat{\lambda}(\omega)}{2}}{1 + 2\alpha(\omega - \omega_k)} \tag{1}$$

$$\omega_k^{n+1} = \frac{\int_0^\infty \omega \left| \hat{\mu}_k^{n+1}(\omega) \right|^2 d\omega}{\int_0^\infty \left| \hat{\mu}_k^{n+1}(\omega) \right|^2 d\omega} \tag{2}$$

Step 3: Update $\hat{\lambda}^{n+1}$ from Equation (3):

$$\hat{\lambda}^{n+1}(\omega) = \hat{\lambda}^n(\omega) + \tau \left( \hat{f}(\omega) - \sum_k \hat{\mu}_k^{n+1}(\omega) \right) \tag{3}$$

Step 4: Given $\varepsilon > 0$ stop the iteration when Equation (4) is satisfied. Otherwise, repeat step 2 to step 4.

$$\varepsilon > \sum_k \frac{||\hat{\mu}_k^{n+1} - \hat{\mu}_k^n||_2^2}{||\hat{\mu}_k^n||_2^2} \tag{4}$$

where the parameters K and $\alpha$ are pre-defined parameters; in this paper $K = 2000$, $\alpha = 4$, chosen with reference to literature [22].

## 3. Recurrent Plot

The specific algorithm flow of recurrent plot [23] is as follows:

(1) The collected time series signals are reconstructed in phase space. Phase space reconstruction is the basis and first step of recurrent plot analysis. For the time series $(x(1), x(2), x(3), \cdots x(n))$ with signal length $N$, the corresponding reconstruction space obtained by time delay is:

$$X(i) = [X(i), X(i+\tau), \cdots X(i+(m-1)\tau)] \tag{5}$$

in which $i = 1, 2, 3, \cdots, N - (m-1)\tau$, $\tau$ is time delay, $m$ is the embedding dimension, and $X(i)$ is used to reconstruct the phase space of the vector. The optimal time delay $\tau$ and the optimal embedding dimension $m$ are obtained by the method of autocorrelation function.

(2) Calculate the Euclidean norm of any two vectors in the reconstructed phase space.

$$D_{i,j}||X_i - X_j|| \quad i, j = 1, 2, 3, \cdots, N - (m-1)\tau \tag{6}$$

(3) Calculate and reconstruct the recursive value of phase space and construct the recursive matrix of phase space.

$$R_{i,j} = \Theta(\varepsilon - D_{i,j}) \quad i, j = 1, 2, 3, \cdots, N - (m-1)\tau \tag{7}$$

$\varepsilon$ is a recursive threshold constant and is usually set as 15% of the standard deviation of the original time series, and $\Theta(\bullet)$ is the Heaviside function:

$$\Theta(x) = \begin{cases} 0 & x \leq 0 \\ 1 & x > 1 \end{cases} \tag{8}$$

Draw a recursive graph of time series signals. When $R_{i,j} = 0$, $(i,j)$ is denoted as a light point; when $R_{i,j} = 1$, $(i,j)$ is denoted as a dark point. The recurrent plot of time series signal is a dot plot drawn in Cartesian coordinate system with time series label $i$ as the horizontal axis and time series label $j$ as the vertical axis. The points and lines in the recursive graph are distributed in the whole graph in a certain law, indicating that there are deterministic components in the signal, which can be used for type identification.

The fault data are encoded and converted into a 2D image by RP after VMD feature extraction, and the encoded image is shown in Figure 1. Figure 1a shows the fault data converted directly by RP to 2D image without VMD extraction, and Figure 1b shows the fault data converted by RP to 2D image after VMD extraction.

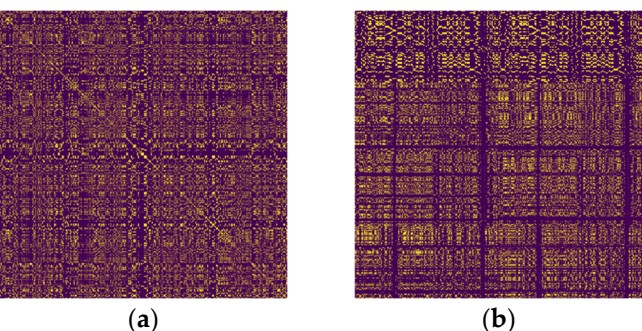

(a)          (b)

**Figure 1.** RP conversion image. (**a**) RP conversion image; (**b**) VMD–RP conversion image.

## 4. Channel Split Residual Network

The training effect of convolutional neural network decreases significantly with the increase of network depth. In order to improve the training effect, the residual neural network (Resnet) proposed by HE [24] can effectively improve the performance of deep

neural network without extra computation by introducing the residual module, whose structure is shown in Figure 2, and the identity operation means no operation is performed.

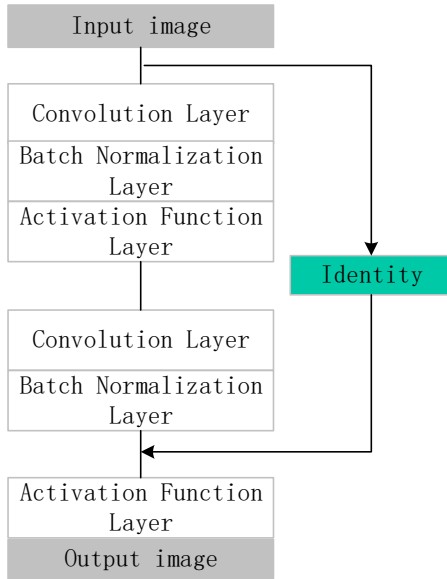

**Figure 2.** Schematic diagram of residual structure.

In order to further improve the feature extraction capability of the residual network and enhance the performance of the deep neural network, this paper proposes a channel split residual network by constructing a channel split (CS) layer instead of the first convolutional layer in the original residual network, and the flow chart of the CS layer is shown in Figure 3. Firstly, the 1D bearing vibration signal is converted into a three-channel 2D picture containing the fault characteristics by the picture conversion method, and then the three-channel picture is input to the network for channel splitting, one of the three channels is selected as the primary operation channel, and the other two channels are secondary operation channels. The channel mixing operation is illustrated in Figure 4. After that, the secondary operation channel is expanded to three channels by convolution. Finally, expanded primary operation channels are stacked *n* times so that the number of channels is the same as the number of channels in the first convolutional layer of the residual network, and then the secondary operation channels are spliced after the primary operation channels.

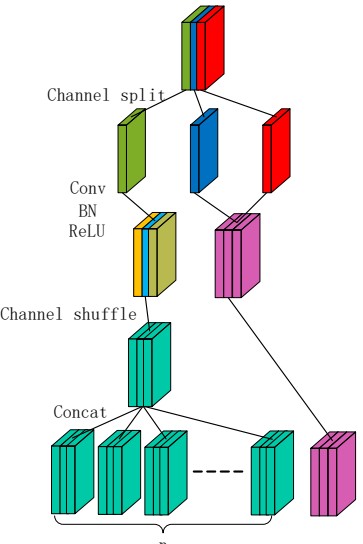

**Figure 3.** CS structure schematic.

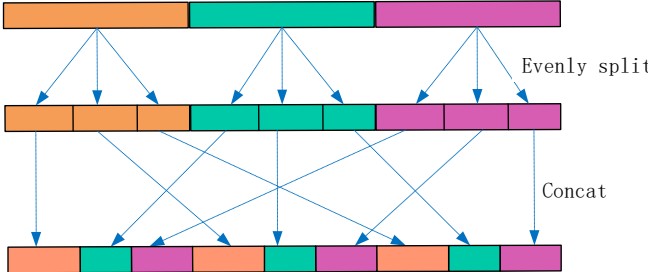

**Figure 4.** Channel shuffle operation diagram.

The final CSRN are shown in Figure 5 where the fault data is input into the CSRN for feature extraction and fault classification after being transformed into a feature image by the RP through VMD feature extraction.

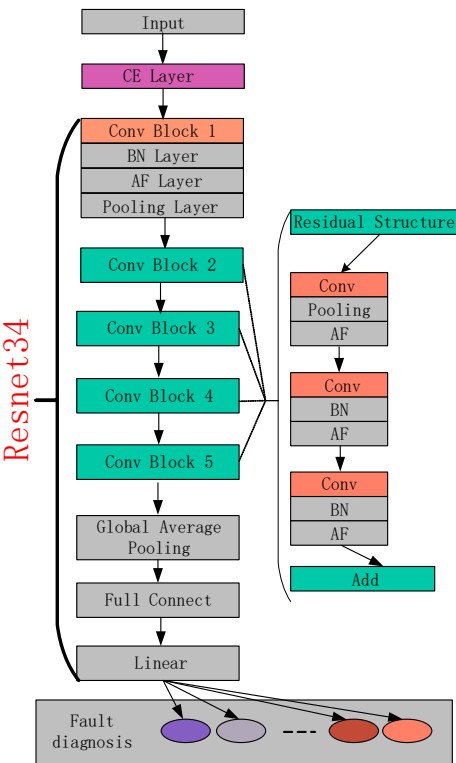

**Figure 5.** CSRN overall flow chart.

## 5. Experimental Data Sets

In order to evaluate the diagnostic accuracy of the proposed VMD–RP–CSRN method for bearing faults, data from the Jiangnan University bearing dataset was selected for experimental validation in this paper. This dataset contains all the operating data of the bearings under different loads.

### 5.1. JNU Bearing Data

JNU bearing data [25] contain bearing faults at 600 r/min, 800 r/min, and 1000 r/min. The bearing operation at different speeds is considered different tasks. For each speed there are four types of data: inner ring failure, outer ring failure, rolling element failure, and normal data, with a sampling frequency of 50 kHz. A total of 10 types of data were selected for each of the three types of failure data and the normal data for the 600 r/min operating condition as the multi-service bearing data set used for the experiments. The specific experimental setup is shown in Figure 6.

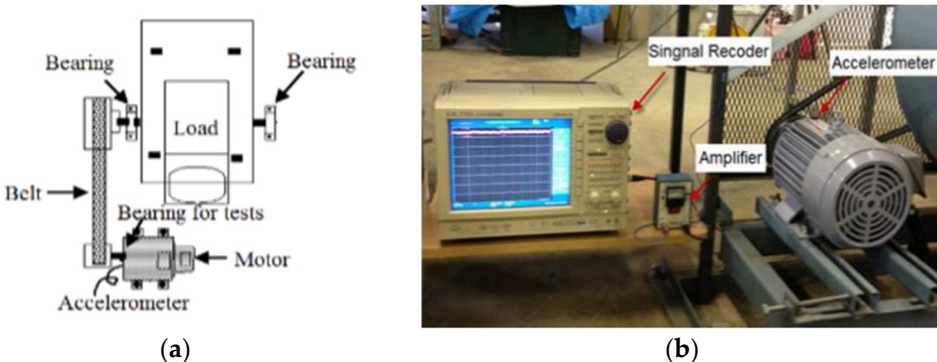

<div align="center">(<b>a</b>)            (<b>b</b>)</div>

**Figure 6.** Experiment setup for the rolling bearing fault diagnosis. (**a**) Illustration of the rotation machinery and (**b**) the motor in the field.

In this dataset, an inductor motor (Mitsubishi SB-JR) used in a centrifugal fan system was used for the faults diagnosis test. The nameplate of the machine was 3.7 kW three-phase induction motors, with Vmax = 220 V, P = 4 pole pairs, and rated speed S = 1800 rpm. Rated slip and frequency were 6.5% and 60 Hz. An accelerometer (PCB MA352A60) with a bandwidth from 5 Hz to 60 kHz and 10 mV/g output was used to measure the vertical vibration signals in the normal, outer-race defect, inner-race defect, and roller element defect states, respectively. The vibration signals measured by the accelerometer were transformed into the signal recorder (Scope Coder DL750) after being magnified by the sensor signal conditioner (PCB ICP Model 480C02). The sampling frequency of the signal measurement was 50 kHz, and the sampling time was 20 s.

### 5.2. Feature Map Generation Method

Firstly, the data on the JNU data bearing set was extracted by VMD, then the extracted data was transformed by RP into a feature map, and the resulting feature images were divided into training and test sets according to a ratio of 7:3 for the transformed data set. The data interception method is shown in Figure 7.

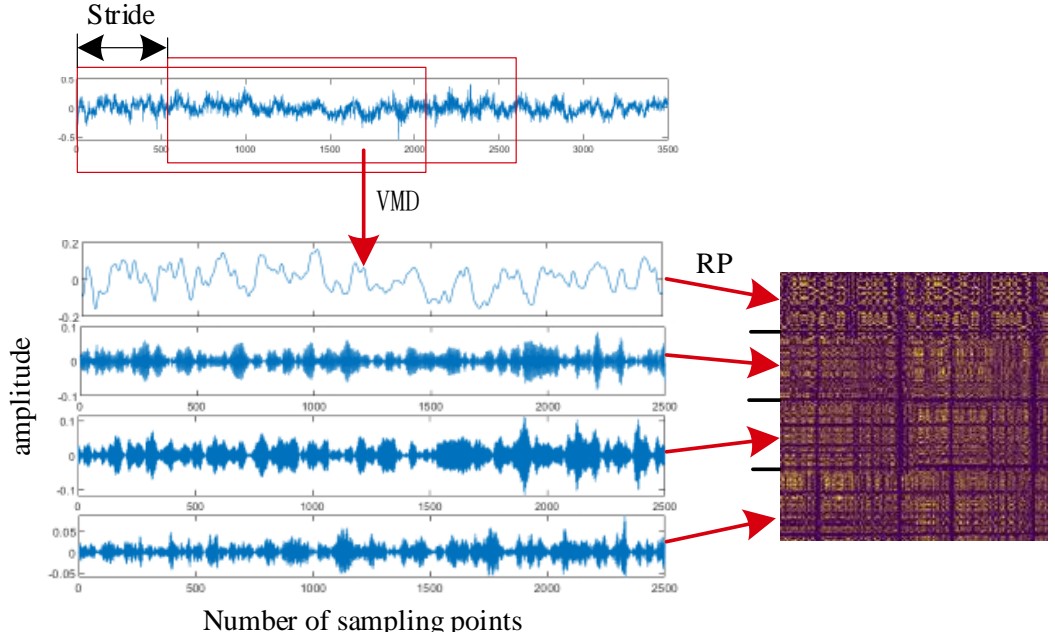

**Figure 7.** Data interception schematic.

In order to fully sample the fault features, the interception length of each data sample was set to 4096 sampling points, and overlap sampling for data expansion was used; the

overlap sampling steps were set to 2048, 1024, and 512 respectively, and divided into training and test sets according to the ratio of 7:3. The specific data set distribution is shown in Table 1.

**Table 1.** Dataset specific division.

|  |  | 2048 | 1024 | 512 |
|---|---|---|---|---|
|  | 600 r/min | 168:75 | 336:150 | 680:292 |
| JN | 800 r/min | 168:75 | 336:150 | 680:292 |
|  | 1000 r/min | 168:75 | 336:150 | 680:292 |

## 6. Experimental Verification

The deep learning framework used for the experiments was Pytorch 1.8.1, and the equipment used was fitted with a RTX3060 graphics card and an i7 11800H CPU. In terms of optimizer, Adam is an excellent optimizer because it can dynamically and smoothly adjust the learning rate of each parameter and can quickly handle the sparse gradient problem of convex functions. In order to eliminate the influence of the optimizer performance on the experimental results of the activation function, we choose Adam as the optimizer for the model used in our experiments. Under the condition of ensuring the normal operation of the program, we sought to make full use of the performance of the hardware by opening four threads at the same time when inputting images. Multi-threading represents that the model can be trained at a faster speed; at the same time, each thread input 16 images at one time, i.e., num-workers = 4, batch-size = 16.

### 6.1. Model Performance Validation

In order to verify the improvement effect of CS layer on the diagnostic accuracy of the model, Jiangnan bearing data containing 10 fault states was selected as the experimental object, and RP and VMD–RP were adopted to convert the fault data and input into the residual network for training and diagnosis, respectively. At this time, the data conversion step was 4096, and the experimental results are shown in Figure 8. When VMD–RP was used to convert the data, the fault diagnosis accuracy of the residual network was 91.1%, which was a 2% improvement compared to the direct conversion of the fault data using RP. When the CS layer was added to the model and the feature maps generated using VMD–RP were input to the model for training and fault classification, the diagnostic accuracy achieved by the model was 92.9% when the main operating channel in the CS layer was channel 2, which was 1.5% and 1.4% higher than when channel 1 and channel 2 were selected as the main operating channels, respectively, and 1.8% higher than when the CS layer was not added. Diagnostic accuracy is the demonstration of the ability of the CS layer to improve the fault diagnostic accuracy of the model. C1, C2, and C3 represent the CSRN models constructed with channels 1, 2, and 3 as the main operational channels for the channel split operation, respectively.

In order to demonstrate the diagnostic capability of VMD–RP–CSRN for bearing faults, the two-dimensional images generated when the data interception step was 512 were taken as the experimental data, used RP and VMD–RP to encode the data respectively and then input them into Resnet for fault diagnosis. The experimental results are shown in Figure 9. For RP, the fault diagnosis accuracy achieved by the model was 96.1%, which is 2.1% higher than that when RP is used as the data conversion method; when the data encoding method was VMD–RP and the diagnosis model was CSRN, the diagnosis accuracy achieved by the model was 97.8%, which is 1.7% higher than that when Resnet is used as the diagnosis model. In addition, to highlight the superiority of the VMD–RP–CSRN diagnostic model, 1DCNN, MTF–CNN [26], and GADF–CNN were used to diagnose the fault data as a comparison, and it can be seen from Figure 9 that the proposed diagnostic method improved the fault diagnosis accuracy by 1.2%, 15.5%, and 13.7%, respectively, compared with the comparison method.

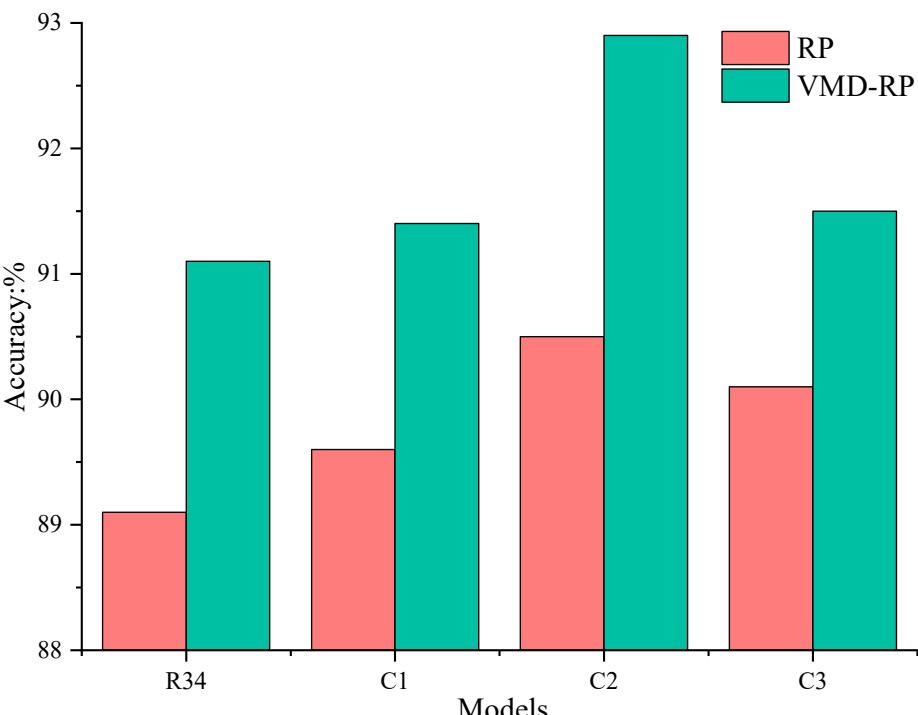

**Figure 8.** Experimental results of model parameter selection.

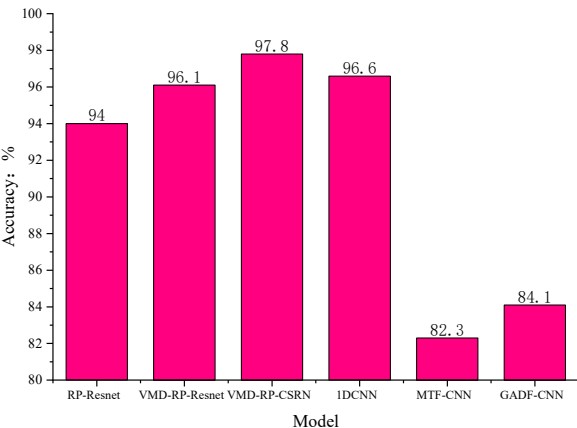

**Figure 9.** Diagnostic performance experiments with the VMD–RP–CSRN model.

### 6.2. Model Noise Immunity Experiments

When bearings are in actual operation, the bearing data is inevitably disturbed by noise during acquisition and has an impact on the fault diagnosis performance of the model. In this paper, Gaussian white noise with different signal-to-noise ratios was added to the original vibration signal to simulate the noise environment in actual operating conditions, and fault diagnosis experiments were conducted using the added noise data to verify the noise immunity of the model. In this paper, the signal-to-noise ratio (SNR) was used as a measure of the noise level, which is defined in Equation (9):

$$SNR = 10lg\left(\frac{Ps}{Pn}\right) \tag{9}$$

where *Ps* is the signal power and *Pn* is the noise power. In this paper, noise of −6 db, −4 db, −2 db, 2 db, 4 db, and 6 db were added to the original data, and the VMD–RP–CSRN is used for fault diagnosis. Due to the excellent performance of 1DCNN, MTF–CNN and GADF–CNN in the field of bearing fault diagnosis, we chose the above methods to

conduct the bearing data after adding noise. The diagnostic experiments were compared with the proposed method in this paper to highlight the performance of the proposed model. The experimental results are given in Table 2. The proposed method achieved an average diagnostic accuracy of 95.52% in six noise cases, which is 11.7%, 34.25%, and 24.6% higher than that of the comparison methods, and the diagnostic accuracy of the proposed method was 93.2% when SNR = −6 db, at which time the highest diagnostic accuracy achieved in the comparison method was 70.7, which was 22.5% lower than that of the paper. The highest diagnostic accuracy achieved in the comparison method was 70.7, which was 22.5% lower than that of the proposed method, thus demonstrating the superiority of the proposed method in terms of noise immunity.

**Table 2.** Experimental results of the VMD–RP–CSRN model for noise.

| SNR | Fault Diagnosis Accuracy: % | | | |
|---|---|---|---|---|
| | **VMD–RP–CSRN** | **1DCNN** | **MTF–CNN** | **GADF–CNN** |
| −6 db | 93.2 | 70.7 | 30.2 | 60.1 |
| −4 db | 94.9 | 76.6 | 32.8 | 63.3 |
| −2 db | 94.9 | 79.8 | 52.2 | 69.8 |
| 2 db | 95.9 | 89.9 | 83.4 | 76.7 |
| 4 db | 97.4 | 92.5 | 84.8 | 76.4 |
| 6 db | 96.8 | 93.4 | 84.2 | 79.2 |
| Mean | 95.52 | 83.82 | 61.27 | 70.92 |

*6.3. Analysis of the Generalisation Performance of the Model for Different Speed Scenarios*

The accuracy of the diagnostic model of diagnosing bearing faults at different speeds is a very important indicator of the diagnostic performance of the model. In order to verify the superiority of the proposed model in diagnostic performance at different rotational speeds, three different rotational speeds were used to build the training set and test set respectively, where A represents the bearing fault data at 600 r/min, B represents the bearing speed of 800 r/min, and C represents the bearing speed of 1000 r/min. A → B represents the training set built with data set A and the test set built with data set B. The training set B was used to construct the test set. In this paper, 1DCNN, MTF–CNN, GADF–CNN, and VMD–RP–CSRN models were used for comparison to highlight the generalization performance of the models, and the experimental results are presented in Table 3. Mean1 represents the mean of the results of constant speed experiments and Mean2 represents the mean of the results of the variable speed experiments.

**Table 3.** Experimental results on the generalization performance of the VMD–RP–CSRN model.

| | Fault Diagnosis Accuracy: % | | | |
|---|---|---|---|---|
| | **VMD–RP–CSRN** | **1DCNN** | **MTF–CNN** | **GADF–CNN** |
| A → A | 97.3 | 93.0 | 94.4 | 91.1 |
| A → B | 95.5 | 89.2 | 80.3 | 69.3 |
| A → C | 96.7 | 84.7 | 78.4 | 73.7 |
| B → A | 95.0 | 91.7 | 68.3 | 77.8 |
| B → B | 98.9 | 96.6 | 97.4 | 92.9 |
| B → C | 96.7 | 96.5 | 87.8 | 80.0 |
| C → A | 96.5 | 88.3 | 66.3 | 81.7 |
| C → B | 96.2 | 94.9 | 95.5 | 85.0 |
| C → C | 99.2 | 94.3 | 96.1 | 91.9 |
| Mean | 96.89 | 92.13 | 84.94 | 82.6 |
| Mean1 | 98.47 | 94.63 | 95.97 | 91.97 |
| Mean2 | 96.10 | 90.88 | 79.43 | 77.92 |

When both the training and test sets had the same speed, i.e., 600 r/min, the proposed method achieved a diagnostic accuracy of 97.3%, which is 4.3%, 2.9%, and 6.2% higher

than that of the comparison methods. In terms of average accuracy, the average diagnostic accuracy of the proposed method is 98.47% with constant speed, which is 3.84%, 2.5%, and 6.5% better than the comparison methods. The diagnostic performance of this method is better than the comparison methods when the speed of the data in the training and test sets is different. With the speed of the training set A and the speed of the test set B, the diagnostic accuracy of this method is 95.5%, which is 6.3% higher than that of the best 1DCNN and 26.2% higher than that of the lowest GADF–CNN. In terms of average accuracy, the method achieved 96.10% diagnostic accuracy at variable speed, which is 5.22%, 16.67%, and 18.18% better than the comparison methods. The analysis of the above comparative experimental results demonstrates that the proposed VMD–RP–CSRN method has better generalization performance in the diagnosis of bearing faults at different speed.

### 6.4. Performance Analysis of the Model at Different Dataset Sizes

The amount of fault data collected during the actual operating conditions of the bearings is usually very limited, resulting in the model not being able to obtain high diagnostic accuracy. Therefore, fault diagnosis performance of the model of small data sets is also an important indicator to evaluate its comprehensive performance. In order to verify the diagnostic performance of the proposed VMD–RP–CSRN model under small data sets, this paper conducted diagnostic experiments on the bearing data set of Jiangnan University with a 50% scale training set and a 10% scale training set for training and a constant scale test set respectively. At 50% of the training set size, there were 340 samples of each fault in the training set, and the number of samples of each fault in the test set taken for the experiment was still not 292; at 10% of the training set size, there were 68 samples of each fault in the training set, and the number of samples of each fault in the test set was still not 292. The results of the diagnostic experiments at 50% and 10% of the training set size are given in Tables 4 and 5, respectively.

**Table 4.** Experimental results of model diagnosis at 50% training set size.

|  | Fault Diagnosis Accuracy: % | | | |
|---|---|---|---|---|
|  | **VMD–RP–CSRN** | **1DCNN** | **MTF–CNN** | **GADF–CNN** |
| A → A | 95.8 | 78.6 | 94.2 | 86.6 |
| A → B | 94.6 | 67.9 | 76.5 | 68.1 |
| A → C | 97.0 | 63.7 | 78.2 | 71.1 |
| B → A | 92.0 | 73.4 | 69.7 | 76.2 |
| B → B | 98.2 | 85.9 | 96.3 | 91.7 |
| B → C | 97.0 | 77.5 | 83.3 | 76.2 |
| C → A | 93.0 | 63.6 | 70.0 | 76.9 |
| C → B | 93.3 | 75.5 | 87.4 | 77.5 |
| C → C | 98.3 | 75.3 | 94.3 | 87.9 |
| Mean | 95.47 | 73.49 | 83.32 | 79.13 |
| Mean1 | 97.43 | 79.93 | 94.93 | 88.73 |
| Mean2 | 94.48 | 70.27 | 77.52 | 74.33 |

At a training set size of 50%, the proposed method achieved average diagnostic accuracy of 95.47%, which was 12.15% better than the MTF–CNN, which has the best diagnostic performance among the comparison methods. When the speed of the training set and the speed of the test set are the same, the proposed method achieved an average diagnostic accuracy of 97.43%, which was 17.5%, 2.5% and 8.7% higher than that of the comparison methods. The proposed method achieved 94.48% average diagnostic accuracy when the training and test sets have different rotational speed, which was 24.21%, 16.96%, and 20.25% better than the comparison methods.

**Table 5.** Experimental results of model diagnosis at 10% training set size.

| | Fault Diagnosis Accuracy: % | | | |
| --- | --- | --- | --- | --- |
| | **VMD–RP–CSRN** | **1DCNN** | **MTF–CNN** | **GADF–CNN** |
| A → A | 92.0 | 60.0 | 80.9 | 76.1 |
| A → B | 80.0 | 56.8 | 75.9 | 71.2 |
| A → C | 80.8 | 47.1 | 77.6 | 67.6 |
| B → A | 82.6 | 58.8 | 67.8 | 74.3 |
| B → B | 90.3 | 69.2 | 92.4 | 79.9 |
| B → C | 86.8 | 51.3 | 82.5 | 64.1 |
| C → A | 83.8 | 52.1 | 62.4 | 71.2 |
| C → B | 87.6 | 58.1 | 86.9 | 68.8 |
| C → C | 96.7 | 53.7 | 88.5 | 73.2 |
| Mean | 86.73 | 56.34 | 79.43 | 71.82 |
| Mean1 | 93.00 | 60.97 | 87.27 | 76.4 |
| Mean2 | 83.60 | 54.03 | 75.52 | 69.53 |

At a training set size of 10%, the proposed method achieved an average diagnostic accuracy of 86.73%, which was 7.3% better than the MTF–CNN with the best diagnostic performance among the comparison methods. When the speed of the training set and the speed of the test set were the same, the proposed method achieved an average diagnostic accuracy of 93.00%, which was 32.03%, 5.73%, and 16.6% higher than that of the comparison methods. When the speed of the training and test sets were different, the proposed method achieved an average diagnostic accuracy of 83.60%, which was 29.57%, 8.08%, and 14.07% higher than that of the comparison methods.

In summary, the proposed method can still achieve very good diagnostic results when the training set sample is reduced, and has better stability than the comparison methods.

## 7. Fault Diagnosis of Variable Speed Bearings

When a bearing works in a variable speed situation with a fault, it is difficult to extract the bearing fault features; the change in speed causes the fault features to change immediately, and the model has low accuracy in classifying the fault. In order to achieve accurate fault diagnosis of variable speed bearings, a fault dataset was generated by converting variable speed bearing fault data through VMD–RP, the training set and test set were divided according to a 7:3 ratio, and the fault classified by CSRN was used for feature extraction and fault classification.

SQV Dataset [27,28]. The experimental data set used the SQ (Spectra Quest) comprehensive mechanical failure simulation test bench to simulate the failure of the outer and inner rings of the motor bearing. The data acquisition time for each test was 15 s, consisting of a complete acceleration/deceleration process from standstill to 3000 rpm and then a steady deceleration to 0. The number of training sets and the number of test sets consisted of the original variable speed data set transformed by VMD–RP to form the feature map data set according to a 7:3 division (Table 6). The experimental results are shown in Table 7. For the original variable speed data, the proposed method achieved 99.3% diagnostic accuracy, which is 39%, 19.9%, and 5.1% better than the comparison methods. In terms of noise immunity, six experiments with different noise additions were conducted in this paper, and the proposed method achieved an average diagnostic accuracy of 98.62% in the six noise experiments, which was 25.1% better than the comparison methods. The average diagnostic accuracy of the proposed method in the six noise experiments was 98.62%, which was 25.82%, 25.12% and 5.74% higher than that of the compared methods. It can be observed that the proposed VMD–RP–CSRN bearing fault diagnosis method has excellent diagnostic performance for variable condition bearing faults, and also has very good noise immunity performance.

**Table 6.** SQV variable speed bearing sample distribution.

| Bearing Condition | Tags | Number of Training Sets | Number of Test Sets |
|---|---|---|---|
| Normal(NC) | 0 | 700 | 300 |
| Inner light (IF_1) | 1 | 700 | 300 |
| InnerModerate (IF_2) | 2 | 700 | 300 |
| InnerSevere (IF_3) | 3 | 700 | 300 |
| Outer light (OF_1) | 4 | 700 | 300 |
| OuterModerate (OF_2) | 5 | 700 | 300 |
| OuterSevere (OF_3) | 6 | 700 | 300 |

**Table 7.** Experimental results on the noise immunity of the model to variable speed datasets.

| SNR | Fault Diagnosis Accuracy: % | | | |
|---|---|---|---|---|
| | VMD–RP–CSRN | 1DCNN | MTF–CNN | GADF–CNN |
| −6 db | 96.9 | 71.1 | 60.8 | 87.7 |
| −4 db | 97.2 | 73.1 | 68.2 | 90.9 |
| −2 db | 99.0 | 74.3 | 70.7 | 92.1 |
| Origin | 99.3 | 60.3 | 79.4 | 94.2 |
| 2 db | 99.5 | 72.8 | 81.9 | 94.6 |
| 4 db | 99.6 | 74.1 | 82.4 | 95.2 |
| 6 db | 99.5 | 71.4 | 77.0 | 96.8 |
| Mean value of noise experiments | 98.62 | 72.80 | 73.50 | 92.88 |

## 8. Conclusions

For traditional fault diagnosis in rolling bearings, actual operating conditions are complex and variable; bearing signal acquisition noise interference and other problems lead to low accuracy of the bearing fault diagnosis method, poor anti-noise results, and other problems. To address these problems, this paper proposes a method based on a VMD–RP–CSRN rolling bearing fault diagnosis model. Firstly, after the initial feature extraction of the bearing signal by VMD, the decomposed signal is converted into a two-dimensional picture with fault features after coding by RP, and then the feature picture is input to CSRN for feature extraction and fault classification. The specific work and related conclusions are as follows:

1. Simple feature extraction of the original bearing signal by VMD is followed by RP coding to generate a 2D picture containing fault features and then input to CSRN for fault feature extraction and classification, ultimately achieving 97.8% diagnostic accuracy, a 1.7% improvement in diagnostic accuracy compared to when the CSRN model was not used, and a 3.8% improvement in diagnostic accuracy compared to the original data coded by RP.

2. Experiments on the generalization performance of the proposed model under different complex operating conditions as well as experiments on the diagnosis of bearing faults under different sizes of training sets show that the VMD–RP–CSRN model has better generalization and stable performance than other algorithms.

3. The proposed model achieves 99.3% diagnostic accuracy in variable speed bearing fault diagnosis experiments, with a minimum improvement of 5.1% compared to the comparison methods, and 98.62% accuracy in noise immunity, with a 5.74% improvement in diagnostic accuracy compared to the comparison methods.

In addition, when using VMD–RP for fault data conversion in this paper, certain values are set for preset parameters in VMD. Although VMD has been verified in relevant literature with excellent performance, its application in bearing fault diagnosis is still uncertain to some extent. In future research, we should try to use particle swarm optimization algorithm to carry out adaptive optimization on VMD, so that VMD can extract fault features more

effectively. At the same time, it is also of certain research significance to transform bearing fault signals into two-dimensional feature images after multi-feature extraction by fusion.

**Author Contributions:** Conceptualization, Y.J. and J.X.; methodology, Y.J. and J.X.; software, J.X.; validation, J.X.; formal analysis, J.X.; investigation, Y.J.; resources, Y.J.; data curation, J.X.; writing—original draft preparation, J.X.; writing—review and editing, Y.J. and J.X.; visualization, J.X.; supervision, Y.J.; project administration, Y.J.; funding acquisition, Y.J. All authors have read and agreed to the published version of the manuscript.

**Funding:** This work was supported by the Key Research and Development Program of Anhui Province under Grant 202104g01020012 and the Research and Development Special Fund for Environmentally Friendly Materials and Occupational Health Research Institute of Anhui University of Science and Technology under Grant ALW2020YF18.

**Data Availability Statement:** The data that support the findings of this study are openly available in http://www.52phm.cn/blog/detail/52 (accessed on 1 December 2022) and https://www.aliyundrive.com/s/TsfYj2UktLR (accessed on 1 December 2022).

**Conflicts of Interest:** The authors declare no conflict of interest.

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
