# Peer review of "VMD–RP–CSRN Based Fault Diagnosis Method for Rolling Bearings"

_electronics, doi:10.3390/electronics11234046_

Round 1

Reviewer 1 Report

Decomposing the mixed signal is very important in the early stage of bearing failure detection. Detecting bearing failure at an early stage prevents failure of the machine in which the bearing is installed, which is also very important from the point of view of preventive maintenance.

In this paper is feature extraction of the bearing signal was carried out by variation modal decomposition (VMD). VMD was then converted into a two-dimensional image with fault features by recurrent plot (RP) coding, and then the feature images were input to a channel split residual network (CSRN) for feature extraction and fault classification.

Figure 7 is small, those markings are not noticeable.

What does the Y axis represent; I assume that the X axis is the frequency in Figure 7.

What is the C1 to C3  model in Figure 8. What does it represent?

What equipment was used during the experimental test (acceleration or velocity sensors, etc., characteristics of the used sensors) and which parameters were used during data collection (sampling frequency, etc)?

Schematic representation of the experimental test is missing.

Where the sensor/s was/were placed?

Since the experimental device is not described, we do not know how many other elements (el.motor, other bearings, shaft, etc. If device is hydrostatic or hydrodynamic, should be emphasized in the paper. Based on this other readers know that the elements of the experimental device will not affect on the results of the tested bearing) of the experimental device influence of the test results.

In the summary line 24 it says:  The experimental results show that the proposed method is at least 1.2% better than the 24 comparison method, and has better noise immunity”. Please indicate the advantages of the proposed VMD-RP-CSRN compared with the other comparison method.

Authors should explain major novelty and contributions of this paper in the introduction section.

From this point of view, I agree to accept the paper with minor revision to publish in the journal as a research paper.

Author Response

Response to Reviewer1 Comments

Dear reviewer, thank you for your patience and attention. According to your comments, we have made some answers and made corresponding modifications in the paper. Please check.

Decomposing the mixed signal is very important in the early stage of bearing failure detection. Detecting bearing failure at an early stage prevents failure of the machine in which the bearing is installed, which is also very important from the point of view of preventive maintenance.

In this paper is feature extraction of the bearing signal was carried out by variation modal decomposition (VMD). VMD was then converted into a two-dimensional image with fault features by recurrent plot (RP) coding, and then the feature images were input to a channel split residual network (CSRN) for feature extraction and fault classification.

  1. Figure 7 is small, those markings are not noticeable.

Response :

We I adjusted the structure of Figure 7 to make the tags more noticeable.

  1. What does the Y axis represent; I assume that the X axis is the frequency in Figure 7.

Response :

In Fig.7, in order to describe the way of data interception, we draw the original data and the data decomposed by VMD, where the X axis represents the number of sampling points, and the Y axis represents the amplitude. After modification, we have marked on the graph.

  1. What is the C1 to C3  model in Figure 8. What does it represent?

Response :

C1, C2, and C3 represent the CSRN models constructed with channels 1, 2, and 3 as the main operational channels for the channel segmentation operation, respectively.

  1. What equipment was used during the experimental test (acceleration or velocity sensors, etc., characteristics of the used sensors) and which parameters were used during data collection (sampling frequency, etc)?

Response :

In this dataset, an inductor motor (Mitsubishi SB-JR) used in a centrifugal fan system is employed for the faults diagnosis test. The nameplate of the machine is 3.7 kW three-phase induction motor, with Vmax = 220V, P = 4 pole pairs, rated speed S = 1,800 rpm. Rated slip and frequency are 6.5% and 60 Hz. an accelerometer (PCB MA352A60) with a bandwidth from 5 Hz to 60 kHz and a 10 mV/g output is used to measure the vertical vibration signals in the normal, outer-race defect, inner-race defect, and roller element defect states, respectively. The vibration signals measured by the accelerometer are transformed into the signal recorder (Scope Coder DL750) after being magnified by the sensor signal conditioner (PCB ICP Model 480C02). The sampling frequency of the signal measurement is 50 kHz, and the sampling time is 20 s.

  1. Schematic representation of the experimental test is missing.

Response :

Additions have been made

Experiment setup for the rolling bearing fault diagnosis. (a)Illustration of the rotation machinery and (b) the motor in the field

  1. Where the sensor/s was/were placed?

Response :

By referring to the relevant papers of the bearing data set of Jiangnan University, we found the schematic diagram of the experimental test. The instruments used are indicated in the drawing.

  1. Since the experimental device is not described, we do not know how many other elements (el.motor, other bearings, shaft, etc. If device is hydrostatic or hydrodynamic, should be emphasized in the paper. Based on this other readers know that the elements of the experimental device will not affect on the results of the tested bearing) of the experimental device influence of the test results.

Response :

We illustrate the effect of other elements on the results of the measured bearings by making references to the paper in which this dataset was first presented.

  1. In the summary line 24 it says: “The experimental results show that the proposed method is at least 1.2% better than the comparison method, and has better noise immunity”. Please indicate the advantages of the proposed VMD-RP-CSRN compared with the other comparison method.

Response :

The proposed bearing fault diagnosis method based on VMD-RP-CSRN combines VMD and RP to retain the fault features in the original signal to the maximum extent and highlight the hidden features in the signal. The proposed channel segmentation operation realizes the extraction of hidden features by selecting the main operating channel of the three-channel feature image, and makes more fault features participate in the feature extraction of the diagnosis model.

  1. Authors should explain major novelty and contributions of this paper in the introduction section.

Response :

  1. Combine VMD feature extraction algorithm with RP image coding to transform feature extraction of fault data into two-dimensional images and enhance the correlation between time series data. On the premise of fully retaining the features contained in the original fault signal, the hidden features in the signal are mined through VMD and expressed through RP.
  2. Build the channel segmentation mechanism, improve the residual network, make full use of the differences of features in different channels of two-dimensional images, and selectively highlight the channels, so as to fully express the hidden feature information in the channels and fully extract the hidden features of RP images.

Thank you again for your patience and attention.

Reviewer 2 Report

The paper is interesting and authors have presented it nicely. There are some o servations as follows:

1. There are some grammatical errors to be improved.

2. Begining of any paper should not start with "as". Please correct the line and rewrite.

3. Objectives are not very clear. Research gap analysis is required.

4. Motivations regarding adoption of VMD feature extraction algo needs better justifications.

5. Most of the variables are not properly defined.

6. "Recurrent Plot are a method" please correct the English.

7. How the the faulty data is encoded and converted into a 2D image? It needs better explanations.

8. Conclusions shjould include limitations and future research directions in better ways.

Author Response

Response to Reviewer2 Comments

Dear reviewer, thank you for your patience and attention. According to your comments, we have made some answers and made corresponding modifications in the paper. Please check.

The paper is interesting and authors have presented it nicely. There are some observations as follows:

  1. There are some grammatical errors to be improved.

Response :

We contacted some colleagues majoring in English to check the article, and she tried her best to modify the article.

  1. Begining of any paper should not start with "as". Please correct the line and rewrite.

Response :

We have corrected this sentence.  “As the core component of rotating machinery, the operating condition of rolling bearings is related to whether the rotating machinery can work safely and stably." It was amended to "Rolling bearing is the core part of rotating machinery, its running condition is related to whether the rotating machinery can work safely and stably."

  1. Objectives are not very clear. Research gap analysis is required.

Response :

In the experimental results shown in Figure 8, we performed some additional experiments to illustrate the performance of the channel slicing operation and its impact on the diagnostic model.

  1. Motivations regarding adoption of VMD feature extraction algo needs better justifications.

Response :

VMD, proposed by Konstantin Dragomiretskiy in 2014, is an adaptive and completely non-recursive approach to modal variational and signal processing, which can adaptively match the optimal center frequency and finite bandwidth of each mode in the search and solution process according to the given number of modal decompositions, and can achieve the effective separation of the intrinsic modal components and the signal The optimal solution of the variational problem is obtained by dividing the frequency domain. In the bearing fault diagnosis, Hongjiang Cui et al. decomposed the bearing vibration signal into a series of intrinsic mode functions by VMD, and then classified the fault features by maximum correlation kurtosis deconvolution, and obtained a better fault diagnosis accuracy. After decomposing the original signal into mode components and dividing the mode matrix into sub-matrices, Chang Liu et al. extracted the local feature information contained in each sub-matrix into singular value vectors using singular value decomposition, constructed the singular value vector matrix corresponding to the current fault state according to the position of each sub-matrix, and finally completed the identification and classification of fault types by the convolutional neural network.

  1. Most of the variables are not properly defined.

Response :

We have added the definitions of some variables to the text.

  1. "Recurrent Plot are a method" please correct the English.

Response :

We have rewritten the passage.

  1. How the the fault data is encoded and converted into a 2D image? It needs better explanations.

Response :

We give a new explanation to the principle of recurrent plot and introduce its principle and drawing process.

  1. Conclusions shjould include limitations and future research directions in better ways.

Response :

We have added limitations and next research directions:

In addition, when using VMD-RP for fault data conversion in this paper, certain values are set for preset parameters in VMD. Although VMD has been verified in relevant literature with excellent performance, its application in bearing fault diagnosis is still uncertain to some extent. In the next research, we should try to use particle swarm optimization algorithm to carry out adaptive optimization on VMD, so that VMD can extract fault features more effectively. At the same time, it is also of certain research significance to transform bearing fault signals into two-dimensional feature images after multi-feature extraction by fusion.

Thank you again for your patience and attention.

Reviewer 3 Report

The manuscript presents an algorithm for bearing fault diagnosis based on some well-known techniques such as VMD and CSRN. However the combination of these methods seems to be unique for this particular application. Therefore the authors should be more clear and better stress the novelty of their work as well as the real need to have one more method for bearing fault diagnosis in addition to many other already developed and successfully tested even with higher accuracy. The following comments are given to further improve the manuscript quality:

1.    Avoid lumping references, e.g. 14-15 and similar. Instead summarize the main contribution of each referenced paper in a separate sentence and/or cite the most recent and/or relevant one. The authors should also consider adding there a couple of more recently published results in the field, e.g.

https://www.mdpi.com/1424-8220/16/3/316 https://www.sciencedirect.com/science/article/abs/pii/S0360544216311525

2.    The authors should more clearly explain why they use VMD for their feature extraction problem and why this method turns out to be more accurate than some others. Have they tested any other?

In overall the contribution of the manuscript is acceptabe but it still needs a minor revision.

Author Response

Response to Reviewer3 Comments

Dear reviewer, thank you for your patience and attention. According to your comments, we have made some answers and made corresponding modifications in the paper. Please check.

  1. The manuscript presents an algorithm for bearing fault diagnosis based on some well-known techniques such as VMD and CSRN. However the combination of these methods seems to be unique for this particular application. Therefore the authors should be more clear and better stress the novelty of their work as well as the real need to have one more method for bearing fault diagnosis in addition to many other already developed and successfully tested even with higher accuracy. The following comments are given to further improve the manuscript quality:

Response :

I have read the two documents you provided and quoted them in the paper. When reading the literature, I found that the methods of the two literatures are from the perspective of practical application, and both have actual experimental equipment for verification. I feel very envious and hope that I can have such a scientific research environment in my future doctoral career, although I am still looking for this opportunity.

“In order to highlight the features in the bearing vibration signals, so that the diagnostic model can make a more accurate diagnosis of bearing faults, some scholars carry out secondary extraction of bearing fault features after feature extraction, and select representative features as the fault classification basis. Jovan Gligorijevic et al. decomposed the bearing vibration signal into several interested sub-bands through wavelet decomposition, and took the standard deviation of the obtained wavelet coefficient as the representative feature to realize the accurate diagnosis of bearing faults. Aleksandar Brkovic et al. made wavelet decomposition of bearing vibration signals, extracted the standard deviation as a measure of average energy and the logarithmic energy entropy as a measure of disorder degree from the interested subbands as representative features, and used the scattering matrix to optimize their dimensions, achieving 100% diagnostic accuracy in the early bearing fault diagnosis.”

  1. The authors should more clearly explain why they use VMD for their feature extraction problem and why this method turns out to be more accurate than some others. Have they tested any other?

Response :

We added the reasons for choosing VMD in the preface. As for other feature extraction methods, compared with VMD, there are some shortcomings, so they have not been considered and experimented. In the future work, we will try a variety of feature extraction methods to fusion and extract fault signals and convert them into feature images, and perform experiments on their fault diagnosis performance.

Thank you again for your patience and attention.
